# A Study of Soil-Borne Fusarium Wilt in Continuous Cropping Chrysanthemum Cultivar ‘Guangyu’ in Henan, China

**DOI:** 10.3390/jof10010014

**Published:** 2023-12-27

**Authors:** Lei Liu, Yaqiong Jin, Miaomiao Chen, Huijuan Lian, Yanyan Liu, Qianxi Yin, Hailei Wang

**Affiliations:** College of Life Sciences, Henan Normal University, Xinxiang 453007, China; l-lei@whu.edu.cn (L.L.);

**Keywords:** cut chrysanthemum, continuous cropping wilt, pathogen invasion, plant physiological response, rhizosphere microorganisms

## Abstract

Cut chrysanthemum, known as a highly favored floral choice globally, experiences a significant decline in production due to continuous cropping. The adverse physiological effects on cut chrysanthemums result from the degradation of a soil’s physical and chemical properties, coupled with the proliferation of pathogens. The “Guangyu” cultivar in Xinxiang, Henan Province, China, has been specifically influenced by these effects. First, the precise pathogen accountable for wilt disease was effectively identified and validated in this study. An analysis was then conducted to examine the invasion pattern of the pathogen and the physiological response of chrysanthemum. Finally, the PacBio platform was employed to investigate the dynamic alterations in the microbial community within the soil rhizosphere by comparing the effects of 7 years of monocropping with the first year. Findings indicated that *Fusarium solani* was the primary causative agent responsible for wilt disease, because it possesses the ability to invade and establish colonies in plant roots, leading to alterations in various physiological parameters of plants. Continuous cropping significantly disturbed the microbial community composition, potentially acting as an additional influential factor in the advancement of wilt.

## 1. Introduction

*Chrysanthemum morifolium*, which belongs to the genus *Chrysanthemum*, is one of the oldest ornamental and medicinal flowers and the second-largest cut flower in the world [1]. It was first cultivated in China and is loved by people all over the world for its attractive colors and shapes. Although the implementation of a noncropping or rotation system has the potential to decrease disease incidence and ensure the high quality of chrysanthemums, the increasing demand for cut chrysanthemums has led to an increase in chrysanthemum cultivation areas, and monocropping is usually the most effective method for maximizing economic benefits. However, during long periods of cultivation, chrysanthemums in continuous cropping areas showed yellowing and wilting leaves, stunted growth, and sharp yield declines. As the cropping time increased, the diseases worsened, resulting in complete failure of harvest in some areas, which seriously affected the economic income of growers and caused huge economic losses [2]. Soil microorganisms, specifically soil-borne pathogens, are believed to be the major cause of this decline in productivity [3].

The rhizosphere represents a dynamic soil region that is regulated by intricate interactions between plants and organisms closely associated with roots. These regions are characterized by a high abundance of microorganisms, often referred to as plants’ second genome [4], which plays a crucial role in promoting plant health and enhancing crop yield. These microorganisms considerably influence mineral element availability, carbon and nitrogen cycling, and the development of soil structure [4,5,6]. The practice of long-term continuous cropping has a profound effect on the structure of soil microbial communities, leading to the emergence of soil-borne diseases and a decline in crop yield. In typical soil conditions, certain native microorganisms possess the ability to suppress the growth of pathogens. However, in the context of continuous cropping soil, pathogens exhibit the capability to swiftly infiltrate and propagate, potentially leading to a reduction in the abundance of beneficial bacteria. Research conducted on the continuous cropping of *Chrysanthemum morifolium* Ramat revealed significant alterations in the abundance of bacteria, disrupting the delicate equilibrium of microorganisms within the rhizosphere soil [7].

The *Fusarium* genus encompasses several economically significant plant-pathogenic species that induce wilt disease in various plants [8], including vegetables, grasses, fruit trees, and flowers. The majority of research efforts predominantly concentrated on *F. oxysporum*, the primary causative agent of fusarium wilt on a global scale [9]. This fungal disease poses a substantial threat to crop production, inflicting severe economic losses, particularly in regions characterized by elevated temperatures and humidity levels. The infected plant in the early stage shows slow growth in comparison with a healthy plant as the disease symptoms begin to appear from the lower leaves. Subsequently, curled and yellow leaves are observed from the bottom to the top, ultimately leading to complete withering and demise. Other species within the *Fusarium* genus, such as *F. solani*, *F. incarnatum*, and *F. falciforme* [10], have been identified as pathogens for chrysanthemums. Wilt disease in chrysanthemums has been reported globally [11], and it has outbreaks in various areas in China, thus progressively emerging as a primary constraint on the advancement of the local chrysanthemum industry.

The advent of third-generation sequencing in recent years, specifically PacBioSMRT sequencing, to enhance the study of soil microbial community structure has had a significant effect on the disciplines of genomics and microbiology [12]. This technology has successfully addressed the limitations of conventional culture methods in identifying microorganisms that are challenging to cultivate or have become inactivated. By enabling a comprehensive exploration of the microflora’s composition in various environments, PacBio sequencing offers a novel and efficient approach to investigating microbial community structure, thereby facilitating substantial advancements in the field of microbial research. Pootakham et al. [13] employed full-length ITS and 16S rRNA genes to categorize and examine the symbiotic algal family and bacterial communities present in Indo-Pacific corals located in the Gulf of Thailand. The findings indicated that environmental factors exerted an influence on the composition of plant structures and the diversity of bacterial communities associated with corals. Furthermore, the study demonstrated the efficacy of PacBioSMAT sequencing in accurately classifying coral-related microbiota at the species level.

Limited comprehensive research has been conducted on the factors that contribute to the occurrence of chrysanthemum’s continuous cropping wilt. Studies pertaining to pathogen infection in chrysanthemums often adopt a descriptive approach, primarily examining pathogen species within host populations rather than exploring the fundamental mechanisms that drive these interactions. So, in the present study, the pathogen that may be associated with *Fusarium* wilt was primarily isolated and purified from the rhizosphere soil in the cultivar “Guangyu”. The invasion pattern of the pathogen was examined, and the physiological response of chrysanthemum plants to stress induced by the pathogen was assessed. Finally, the PacBio platform was utilized to analyze complete 16S rRNA and ITS sequences to investigate the attributes of microbial communities in the rhizosphere of soil from the local cut chrysanthemums that have been subjected to continuous cropping for a duration of 7 years and in those of soil from initial cropping. This study aimed to offer a comprehensive understanding of the potential mechanisms that underlie wilt disease in “Guangyu”. The findings could contribute to the establishment of a vital theoretical basis for the sustainable advancement of diverse crop varieties.

## 2. Materials and Methods

### 2.1. Field Investigation and Soil Sample Collection

A field investigation on *Fusarium* wilt was conducted in a cut chrysanthemum planting base located in Xinxiang, Henan Province, China (35°24′ N, 114°55′ E). There were five districts for each continuous cropping and healthy cropping field (first-year cropping) of chrysanthemums, from which 20 chrysanthemum samples were randomly selected. The height and leaf width of each plant were measured. Several chrysanthemum samples exhibiting severe disease symptoms were randomly selected from each district, and their root and stem characteristics were compared with those of healthy plants. Samples of rhizosphere soil were collected and analyzed from 7 years of continuous cropping and first-year cropping fields. Five sampling points in each field were selected, and the collected samples were subsequently merged. The samples were promptly placed into sterile bags, transported to the laboratory under low-temperature conditions, and stored at −80 °C [14].

### 2.2. Isolation and Identification of Pathogen

By using the dilution-plate method, fungi were isolated from soil samples collected from a *Fusarium* wilt field. The strains were isolated and purified by single spore isolation, inoculated into a PDA solid medium, and incubated at a temperature of 30 °C for 3–7 days [15]. When the hyphae were fully covered on a 90 mm plate, the colony morphology, color, texture, and growth rate were carefully observed and recorded. The purified fungi were subsequently cultured in PDA for 7 days. Afterward, the surface conidia were washed with aseptic water and filtered using four layers of lens paper. Finally, 20 μL of the filtered solution was placed under a light microscope (HFX-IIA, Nikon, Tokyo, Japan) to observe the morphology of the conidia.

The fungal genomic DNA was extracted from fresh fungal cultures following the methodology described by Al-Sadi et al. [16]. The rDNA-ITS region was amplified using the universal primers ITS1 (5′-TCCGTAGGTGAACCTGCGG-3′) and ITS4 (5′-TCCTCCGCTTATTGATATGC-3′) as outlined by White et al. [17]. Subsequently, the amplified DNA fragments were purified and ligated into the pMD19-T vector. The resulting constructs were then submitted to Sangon Biotech Co., Ltd. (Shanghai, China) for sequencing. The obtained sequencing results were compared and analyzed using BLAST program on the NCBI website (http://blast.st-va.ncbi.nlm.nih.gov/Blast.cgi (accessed on 21 August 2022)), and a phylogenetic tree was constructed using Clustal X software version 1.83 [18] and MEGA version 6.0.6 [19].

### 2.3. Pathogenicity Test

Healthy chrysanthemum plants (“Guangyu” cultivar) that remained asymptomatic after three generations of consecutive cutting and transplanting in an indoor environment were chosen as the inoculated hosts [20]. The method described by Getha et al. [21] was employed to determine which *Fusarium* species were responsible for inducing plant disease. In the treatment group, each plant was subjected to a fungal spore suspension of 20 mL at a concentration of 10^6^/mL. The control group (CK) was treated with sterile water. The treated plants were then placed in an illuminating incubator and cultured at 30 °C in a 14 h light/10 h dark cycle. The plants were observed until the appearance of disease spots. Subsequently, the diseased stem tissues were collected, and the pathogen was isolated and purified from the tissues. Finally, the isolated fungal specimen was compared with the inoculated pathogen to determine whether it was the causative agent of chrysanthemum wilt.

### 2.4. Observation of Colonization Process and Disease Assessment

Root dipping was employed to investigate the pathway of pathogen invasion [22]. The treatment group was immersed in the prepared spore solution for 30 min, whereas CK was immersed in aseptic water for the same time. The morbidity and mortality rates of the chrysanthemums were then diligently monitored and recorded every 10 days. The disease index (DI) was assessed on the basis of a 0–4 grade by referring to Alkher et al. [23]. Grade 0 indicated the absence of symptoms in leaves; grade 1 denoted the presence of a single leaf exhibiting yellowing or curling at the basal region < 30%; grade 2 indicated yellowing, curling, or wilting of approximately 30–50% leaves, accompanied by a slight reduction in plant height; grade 3 denoted yellowing, curling, or wilting of approximately 50–75% leaves, resulting in leaf abscission; and grade 4 indicated 75–100% yellowing, curling, or wilting of all leaves or the demise of the plant. The disease grade for each plant was documented at 10 and 20 days post inoculation (dpi). Subsequently, DI was computed using the following formula: disease index = Σ (number of diseased leaves at each grade × corresponding grade)/(total number of leaves examined × highest grade).

The root tissues of chrysanthemum were collected at specific time intervals (12 h, 5 days, 10 days, 15 days, and 20 days). For each period, 20 fresh root segments were carefully selected and prepared for scanning electron microscopy (SEM) in accordance with the method described by Boamah et al. [24]. The root samples were subjected to WGA-AF488 and PI co-staining to observe the colonization of pathogens in the root system by laser confocal scanning microscopy (LCSM) [25].

### 2.5. Plant-Cell-Wall-Degrading Enzyme (CWDE) Activity of Pathogen

The enzymatic activity responsible for degrading the cell wall of plants by the pathogen was assessed using the 3,5-dinitrosalicylic acid method [26], specifically measuring the activities of cellulase (CX), β-glucosidase (βG), pectin methylgalactuionase (PMG), polygalacturonase (PG), and xylanase.

### 2.6. Physiological Response of Plants after Infection by Pathogen

The concentrations of photosynthetic pigment, including carotenoids and chlorophylls (chla, chlb), soluble sugars, and soluble proteins, in plants were quantified using spectrophotometry [27], anthrone colorimetry [28], and the Coomassie bright blue method [29], respectively. The levels of ash content, potassium (K), phosphorus (P), and calcium (Ca) in leaves were measured using inductively coupled plasma–mass spectrometry [30].

The content of H_2_O_2_ was measured by following Velikova et al. [31]. The activities of superoxide dismutase (SOD), catalase (CAT), peroxidase (POD), and malondialdehyde (MDA) were determined using kits (Grace Biotechnology, Suzhou, China) in accordance with the manufacturer’s instructions [32]. The activity of polyphenol oxidase (PPO), phenylalanine ammonialyase (PAL), and chitinase (CHI) was determined using PPO, PAL, and CHI assay kits (Nanjing JianCheng Bioengineering Institute, Nanjing, China) on the basis of the manufacturer’s protocols [33].

The content of salicylic acid (SA) and jasmonic acid (JA) in the supernatant was measured using SA and JA ELISA kits (RXJ1401587PL, Quanzhou Ruixin Biological Technology Co., Ltd., Quanzhou, China) [34].

### 2.7. Soil DNA Extraction, PCR Amplification, and Sequencing

The soil samples from mixed continuous cropping were subjected to parallel sequencing and labeled as LZ1861a-a,b,c, whereas the healthy soil samples were labeled as XT1861a-a,b,c. The DNA extraction process for fungi and bacteria in each soil sample followed the standard protocol of the OMEGA DNA isolation kit (Omega, Honolulu, HI, USA). The quality of the extracted DNA was assessed using 1% agarose gels. The purity of DNA was assessed using a NanoDrop One UV-Vis spectrophotometer (Thermo Fisher Scientific, Waltham, MA, USA). Subsequently, the DNA concentration was determined using a Qubit 4.0 fluorometer (Invitrogen, Waltham, MA, USA). The full length of the 16s rRNA gene was amplified using primers 27F/1541R [35]. The ITS1/ITS4 primers were used to amplify the full-length ITS rRNA gene [17]. Following purification, the amplified product was submitted to Grandomics Co., Ltd. (Wuhan, China) for DNA sequencing using the PacBio RS II sequencer (Pacific Biosciences, Menlo Park, CA, USA).

### 2.8. Microbiome Profiling

The initial dataset was partitioned on the basis of barcode information by using SMRT Portal version 2.3.0, resulting in the acquisition of high-quality circular consensus sequencing sequences. Subsequently, the split sequences were filtered to eliminate extraneous data, including fragments outside the length range of 1400–1600 bp, reads containing “N” bases, reads containing homopolymers exceeding six base pairs, and sequences with an average mass below 90. The set of filtered high-quality data obtained was subjected to a query against the GenBank nonredundant nucleotide database (nt) to identify taxa in the National Center for Biotechnology Information (NCBI). Subsequently, it was clustered using the operational taxonomic unit (OTU) methodology. The representative sequences of OTU at 97% similarity level were subjected to classification and analysis using the UCLUST algorithm in QIIME2 software (version 2020.6). The species present in each sample were classified and quantified at various taxonomic levels, including boundary, phylum, class, order, family, genus, and species. The relative abundance of species at the gate and genus levels was visualized using R software (version 3.3.2). Bacterial and fungal communities were compared against the Silva (version 138.1) and Unite (version 8.2) databases, respectively. The α diversity, β diversity, and species differences among the samples were assessed using QIIME and R software.

### 2.9. Statistical Analysis

Statistical analysis was performed using Excel (version 2016) and GraphPad Prism (version 8.0.2) for Windows. The data were expressed as mean ± standard error of the mean and assessed through two-way ANOVA of three biological replicates, followed by the least significant difference test. Statistical significance was determined at *p* < 0.05, *p* < 0.01, and *p* < 0.001.

## 3. Results

### 3.1. Chrysanthemum Disease Symptoms in the Field

Negative physiological characteristics were observed in chrysanthemum plants following continuous cropping. Initially, the lower leaves exhibited wilting and yellowing, whereas no evident browning was observed at the stem base. As the disease progressed, the plants exhibited stunted growth, blackening and rotting of the diseased roots, and increased susceptibility to uprooting. Ultimately, the entire plant withered and perished (Figure 1A,B). According to the statistical analysis conducted on healthy and diseased plants in the field (Figure 1C), the leaf width and plant height of diseased plants were significantly lower than those of healthy plants (*p* < 0.001). This observation suggested that the presence of Fusarium wilt disease had a substantial detrimental effect on the overall quality of chrysanthemum plants.

### 3.2. Isolation and Identification of Fusarium Pathogens

A sample of rhizosphere soil from wilted chrysanthemum plants was taken, and a total of 53 strains of fungi were isolated and purified using gradient dilution plate culture. Among these fungi, 37 were initially identified as Fusarium species based on morphological observation. The 37 strains exhibited two distinct colony forms, referred to as type A and type B. Type A colonies displayed an intermediate raised villous structure, with a neat edge and dense aerial mycelium (Figure 2A,B). The color of the colonies changed from white to light yellow, and wheel lines appeared after continued cultivation. After 7 days of cultivation, the entire plate became completely covered with colonies. Two distinct types of conidia were produced by type A fungus. On the one hand, the large conidia exhibited a sickle-shaped morphology, which was colorless and transparent, and possessed 2–4 septa. They measured approximately 15–20 µm in length and 2–3 µm in width. On the other hand, the small conidia were oval-shaped, possessed 0–1 septum, and had dimensions of 4–5 µm in length and 2–3 µm in width (Figure 2C,D). The type B colony displayed protruding villi in the central region, although its colony edges were irregular. The aerial hyphae of this colony were longer than those of type A hyphae. As the culture time progressed, the colony gradually transitioned in color from red to purple. The colony lacked a ring pattern and exhibited slower growth than type A. After 10 days of culture, the entire plate became covered (Figure 2E,F). The conidia of the type B fungus were characterized by their small, rounded, and colorless morphology, with 0–1 septum and a size ranging from 2 μm to 9.5 μm. Notably, no large conidia were visible (Figure 2G,H).

The amplified rDNA-ITS sequences of 37 strain fragments had a length of approximately 600 bp. Subsequently, the fragment was ligated to pMD19-T and sent to Shenggong Bio Co., Ltd. (Shanghai, China) for sequencing. Comparison of the sequences with the NCBI Blast database showed that among the 37 strains, the sequence of 16 type B strains exhibited complete consistency with *F. oxysporum*, with a homology of 100% (designated as Fo). Conversely, the remaining 21 type A strains displayed a homology of 99.8% with *F. solani* in their ITS sequences (designated as Fs). The phylogenetic tree further proved the isolated strains were conclusively identified as *F. oxysporum* and *F. solani* (Figure 2I).

### 3.3. Pathogenicity Determination of Pathogen

The strains of Fo and Fs were subsequently reintroduced to healthy plants under identical growth conditions. The findings indicated that chrysanthemums in the aseptic water treatment group (CK, Appendix A) and Fo treatment group (Appendix A) exhibited normal growth without any signs of disease. Conversely, the chrysanthemums in the Fs treatment group displayed symptoms consistent with those observed in naturally occurring chrysanthemum wilt in the field (Appendix A). Specifically, the lower leaves began to curl and yellow from 30 dpi, with an incidence rate exceeding 80%. The pathogen was subsequently reisolated from the stem base of the affected plant, and the colony and conidia were consistent with those of the previously isolated.

### 3.4. Invasion and Colonization of F. solani in Chrysanthemum Root

The plant cell wall serves as the initial barrier against pathogen invasion. Pathogens are capable of generating certain substances that break down polymers, including pectin and cellulose, within the plant cell wall to disrupt the structural integrity of plant cell tissue and reduce cell adhesion. These substances were observed to be CWDEs, such as Cx, pectinase, hemicellulase, and others. The secretion of CWDEs by pathogens plays a vital role in the process of infection. According to the data presented in Appendix A, *F. solani* exhibited the capacity to synthesize a range of plant CWDEs, including Cx, βG, PG, PMG, and xylanase. These enzymes may play a crucial role in enabling *F. solani* to penetrate plant tissue [36].

The characteristic symptoms observed in potted chrysanthemum plants affected by *F. solani* were as follows: During the initial phase (0–10 days), the lower leaves gradually exhibited curling, indicating that the disease severity reached grade 1. In the intermediate phase (10–15 days), approximately 30% of the lower leaves display yellowing and wilting, accompanied by browning of the roots, indicating a disease severity of grade 2. In the later stage of the disease (15–20 days), a significant proportion of leaves (50–75%) exhibited yellowing and withering at the bottom, and individual leaves gradually turned brown. The base of the stem displayed a dark brown color, and the disease severity reached grade 3. Following a growth period of 20 days, the entire plant’s leaves assumed a yellow-brown hue, making it susceptible to easy uprooting. Furthermore, the roots underwent rotting and blackening, accompanied by an unpleasant odor. Ultimately, the entire plant withered and perished, resulting in a disease severity of grade 4. The incidence and incidence index statistics at 10 and 20 dpi are shown in Appendix A. The symptoms exhibited by the *F. solani* treatment group closely resembled the naturally occurring symptoms of chrysanthemum wilt observed in the field. Conversely, the potted chrysanthemum plants in CK remained symptom-free and exhibited healthy growth (Figure 3).

The pathway by which *F. solani* infiltrated the plant root is depicted in Figure 4. Following a 12 h invasion period, a limited quantity of conidium adhered to the root hair and intercellular space, with some conidium initiating germination to generate bud tubes (Figure 4B). However, no invasive structures were observed. At 1 dpi, the pathogen’s hyphae that attached to the root surface began elongating and intertwining (Figure 4D), although they remained confined to the cell gaps to seek opportunities for invasion. At 3 dpi, a notable increase was observed in pathogen abundance within the root system, accompanied by the infiltration of certain hyphae through the intercellular space and the attachment of numerous conidia in intercellular space (Figure 4F). At 5 dpi, pathogen propagation and subsequent conidia production commenced on the root surface, coinciding with the partial destruction of cellular structures (Figure 4H). At 10 dpi, the plant had transitioned into the initial phase of the disease, characterized by the presence of hyphae and conidia covering the surface of the root system, leading to a substantial proliferation of the pathogen, and the morphology of the root cells appeared abnormal (Figure 4J–L). By contrast, the CK root consistently exhibited intact and well-organized cell morphology, with cells appearing full, smooth, and closely aligned (Figure 4A,C,E,G,I).

The invasion and colonization of pathogens in plant roots were visualized using LCSM, as depicted in Figure 5. In CK, the plant cells were consistently arranged in a dense and orderly manner, with no evidence of pathogen hyphae or conidia observed within the tissue (Figure 5A–Q). Conversely, in the pathogen treatment group (Fs), a limited number of pathogen conidium was observed to adhere to the root hairs and aggregate in the intercellular space at 12 h post inoculation (Figure 5F,K,P), whereas the cellular structure remained intact. At 1 dpi, the conidium that accumulated in close proximity to and within the root hairs initiated the process of hyphal formation (Figure 5G,L,Q). Consequently, the mycelium structure became discernible within the plant roots, exhibiting growth and extension along the intercellular space. However, the colonization of pathogens within the roots remained limited. At 3 dpi, the hyphae within the root continued to proliferate along the intercellular space, and the germinated conidia that had attached themselves externally to the root commenced invasion through the intercellular spaces, with observable apical growth ends (Figure 5H,M,R). At 5 dpi, the pathogens present in the root system transitioned into the asexual reproductive phase (Figure 5N–P), resulting in the generation of a significant quantity of conidia. Subsequently, these pathogens initiated the upward transportation of conidium through the intercellular space and vascular bundles (Figure 5S). By 10 dpi, the hyphae had successfully colonized the entire root system of the chrysanthemum plant (Figure 5J,O,T). Thus, the aboveground leaves of the chrysanthemum plant began to exhibit signs of water loss and subsequent shrinkage.

### 3.5. Chrysanthemum’s Physiological Response to F. solani Inoculation

#### 3.5.1. Effects on Plant Nutrition and Growth

In the intermediate stage of infection, which occurred 15 dpi, a substantial decline in growth indices of plants, encompassing fresh weight, plant height, and photosynthetic pigments (chla, chlb, and carotenoids) was observed in the treatment group (Fs). The ash content in tissues exhibited a notable increase in comparison with that in CK (Figure 6A). The contents of inorganic P, K, and Ca began to accumulate and reached the peak during the early stage of pathogen infection (0–10 dpi) and subsequently decreased. Furthermore, an increase in soluble sugar and protein concentrations was observed, but as the severity of the plant disease intensified, the levels of inorganic substances and nutrients gradually diminished in the Fs group. Conversely, the inorganic substances and nutrients in CK did not show any noteworthy alterations throughout the experiment (Figure 6C).

#### 3.5.2. Oxidative Stress on Plants

Within 5 dpi of *F. solani* (Fs), the H_2_O_2_ content in the infected tissue exhibited minimal variation. However, after 5 dpi, a rapid increase in H_2_O_2_ levels was observed, peaking at 10 dpi. This increase was significantly higher than that in CK (*p* < 0.001), suggesting the occurrence of oxidative stress in plants. Consequently, upon pathogen infection and the subsequent buildup of H_2_O_2_, the concentration of malondialdehyde (MDA) in plants exhibited a rapid increase starting at 5 dpi, and the increase was sustained and reached its peak at 10 dpi, remaining consistently high thereafter (Figure 6B).

#### 3.5.3. Changes in Defense-Related Enzyme Activity

The antioxidant enzyme activity in plant leaves during infection was measured, and the finding is presented in Figure 6E. Within 3 dpi, the CAT activity exhibited a rapid increase, reaching its peak, and then gradually declining. However, even after 7 dpi, the CAT activity in Fs remained higher (*p* < 0.01) than that in CK. Subsequently, a further decline was observed, with no discernible difference compared with CK during the middle and late stages of infection. The POD activity in CK exhibited an initial increase from 0 dpi to 5 dpi, followed by a subsequent decline until reaching a point where it did not significantly differ from that in CK at 15 dpi. Conversely, the SOD activity in the plants did not display any significant differences between the CK and Fs groups (Figure 6E), suggesting that it may not have been activated following pathogen inoculation. The activation of defense enzymes in leaves varied following pathogen inoculation, as depicted in Figure 6D. The findings indicated that the activity of PPO in Fs started increasing from 0 dpi, reaching its peak at 3 dpi, and subsequently declining. However, it remained higher than that in CK until 7 dpi (*p* < 0.01). The PAL activity exhibited a progressive increase from 3 dpi, corresponding to the gradual increase in infection, and it was higher than that in CK (*p* < 0.01). The CHI activity displayed an ascending pattern from 5 dpi, reaching its zenith on day 10 dpi and subsequently declining.

#### 3.5.4. Plant Hormone Level

The data presented in Figure 6F demonstrated that *F. solani* infection in chrysanthemum roots induced the synthesis of endogenous hormones JA and SA. The concentration of JA exhibited an initial increase upon infection, reaching its peak at 5 dpi, which was higher than that in CK (*p* < 0.01). However, this increase was transient. Conversely, the concentration of SA in Fs showed a rapid increase from the early stages of infection and remained significantly higher than that in CK (*p* < 0.001) until 7 dpi. Subsequently, it gradually decreased but remained higher than that in CK.

### 3.6. Rhizosphere Microbial Community Diversity Changes during Continuous Monocropping

#### 3.6.1. Changes in Bacterial Community Composition

Chao1, Richness, Shannon, and ACE indices were utilized to evaluate the abundance and diversity of bacteria in soil samples. The results reveal a significantly higher bacterial species abundance and diversity in soil samples from continuous cropping compared with healthy soil. The sequencing details and α diversity index of the samples are presented in Appendix A. The sparse and Shannon diversity curves are depicted in Appendix A. They offered an evaluation of the sequencing quantity for each sample. The presented curves provided evidence that the sequencing depth employed facilitated a thorough depiction of bacterial diversity within the sample while enabling the identification of a substantial proportion of the microbial composition.

Figure 7A presents the bacterial community structure and dominant species at the phylum level for continuous cropping soil and healthy soil. The phyla Planctomycetota, Bacteroidota, Proteobacteria, Acidobacteriota, and Chloroflexi exhibited the highest relative abundance of bacterial communities in continuous cropping soil (LZ1861a-a,b,c) and healthy soil (XT1861 a-a,b,c), ranging from 19.73% to 22.82%, 16.10% to 17.08%, 13.01% to 16.54%, 6.98% to 11.23%, and 8.48% to 10.28%, respectively. Collectively, these phyla accounted for 64.30–77.95% of the total bacterial community in the soil. The relative abundance of phyla, such as Patescibacteria, Verrucomicrobiota, and Methylomirabiota, in continuous cropping soil exhibited a statistically significant increase compared with those in healthy soil, with increases of 44.32%, 48.62%, and 33.33%, respectively. Conversely, the relative abundance of Actinomycetota, Cyanobacteria, and Firmicutes in continuous cropping soil significantly decreased by 54.79%, 97.97%, and 85.47%, respectively, compared with that in healthy soil. At the genus level, the differences in bacterial community structure between continuous cropping soil and healthy soil are shown in Figure 7B. In comparison with the bacterial community observed in healthy soil, the relative abundance of beneficial bacteria, namely *Luteimonas*, *Nitrosospira*, *Pirellula*, *Terrimonas*, *Actinomyces*, and *Steroidobacteria*, exhibited a significant decrease in continuous cropping soil. Conversely, numerous unclassified genera, such as Bacteroidota *env.OPS17*, and unknown genera like *AKYG587* and *Rokubacteriales* in Planctomycetota experienced a substantial increase.

Principal coordinate analysis (PCoA) was employed to investigate dissimilarities in the composition of sample communities. Additionally, hierarchical clustering analysis was conducted to construct a dendrogram (Appendix A) that visually represents the influence of continuous cropping on the structure of soil bacteria communities. The findings revealed that the distances between all items on the coordinate axis of the continuous cropping soil samples (LZ1861 a-a,b,c) were greater than those of the healthy soil samples (XT1861 a-a,b,c), indicating that perennial continuous cropping exerted a more substantial effect on the bacterial community structure within the soil. LEfSE analysis was conducted on the sequencing samples, and the communities or species that exhibited noteworthy disparities in the samples are depicted in Appendix A. The findings indicated that the species belonging to Planctomycetes, Actinobacteria, Bacilli, Chitinophagales, and Cyanobacteria in healthy soil and the species in Phycisphaerae, Patescibacteria, Pedosphaeraceae, and Geosphere bacteria in continuous cropping soil were the primary contributors to significant variations in community structure.

#### 3.6.2. Changes in the Fungal Community Composition

The Chao1, Richness, Shannon, and ACE indices were measured to evaluate fungal abundance and diversity in soil samples. The findings indicated a significant decrease in species abundance and diversity of fungi in soil samples subjected to continuous cropping compared with healthy soil. The sequencing information and α diversity index of the samples are presented in Appendix A. The sparse and Shannon diversity curves of continuous cropping soil and healthy soil samples demonstrated that the fungal diversity in the samples was adequately represented at this sequencing depth (Appendix A), and it met the necessary prerequisites for subsequent bioinformatics analysis.

The fungal community structure and dominant species of continuous cropping soil and healthy soil are depicted at the phylum level in Figure 8A. Both soils exhibited three main phyla, namely Ascomycota, Basidiomycota, and Chytridiomycota, with Ascomycota being the most abundant phylum in the soil samples. Notably, the relative abundance of Ascomycota significantly decreased in continuous cropping soil samples, whereas the relative abundance of Basidiomycota was significantly greater than that in healthy soil samples. Significant differences in the fungal community composition were observed at the genus taxonomic level between the soil subjected to continuous cropping and the healthy soil (Figure 8B). In comparison with the fungal community found in healthy soil, the continuous cropping soil exhibited a significant decrease in the relative abundance of beneficial species belonging to *Microdochium*, *Aspergillus*, *Ceratobasidium*, and *Torula*. Conversely, a noteworthy increase was found in the relative abundance of species affiliated with *Ascobolus*, *Myriococcum*, *Rhizophlyctis*, *Lodophhanus*, and *Piskurozyma*. The findings indicated that continuous cropping practices had a pronounced influence on the composition and distribution of fungal communities within the soil. Notably, the pathogen responsible for the occurrence of chrysanthemum wilt in the local area was identified as *F. solani*, which differs from previous reports. Consequently, a detailed analysis of the *Fusarium* community structure was conducted at the species level, and the corresponding results are presented in Figure 8C. The findings indicated that the practice of continuous cropping resulted in a notable increase in the relative abundance of *F. solani*, aligning with previous research outcomes [3,11]. Furthermore, the previously documented pathogen *F. oxysporum* constituted less than 1% of the *Fusarium* present in the local soil samples, potentially attributed to variances in geographical conditions.

The fungal community composition in the samples was examined using PCoA (Appendix A). The findings indicated that perennial continuous cropping had a substantial impact on the fungal community structure in the soil. LEfSE analysis was conducted to further analyze the sequencing samples, revealing the fungal species that exhibited significant differences among the samples (Appendix A). In healthy soils, the community structure displayed notable variations primarily due to the presence of Ascomycota, Sordariomycetes, Dothideomycetes, Xylariales, Microdochium, and Aspergillacea. Conversely, Chytridiomycota, Pezizales, Atheliales, Botryoderma, and Cantharellus were found to be predominant in soils subjected to continuous cropping.

## 4. Discussion

Various pathogens, including *F. incarnatum*, *Dickeya chrysanthemi*, *Rhizoctonia solani*, *Erwinia chrysanthemi*, and *F. oxysporum*, have been identified as potential causes of wilt in chrysanthemums [37]. Through pathogenicity detection and verification, the results of the present research demonstrate that *F. solani* displayed pathogenicity, with pathogenic characteristics consistent with those observed in the natural field. Therefore, it was recognized as the primary pathogen accountable for chrysanthemum wilt in “Guangyu”. However, in another study, *F. oxysporum* and *F. solani* were proven to be the major pathogenic species in continuously monocropped chrysanthemums [3], illustrating that different cultivars of chrysanthemum may exhibit varying degrees of susceptibility to distinct species within the *Fusarium* genus. The strain’s capacity to produce Cx, βG, PG, PMG, and xylanase facilitated fungal penetration of the plant cell wall. Furthermore, the infection induced by this pathogen exerted a substantial inhibitory effect on the growth and metabolic activities of the afflicted plants. Soluble proteins and sugars have been proven to be valuable indicators for evaluating the physiological metabolism of plant cells. The presence of pathogen infection could trigger the defense response in plants, resulting in the production of pathogenesis-related proteins. Sugars play a crucial role in the material and energy metabolism of plants. An increase in soluble sugar levels facilitates the maintenance of cell osmotic pressure, the augmentation of metabolic capabilities, and adaptation to stressful conditions. The findings of this study indicated that the infection of chrysanthemum by *F.solani* resulted in a significant increase in the levels of soluble protein and sugar in the leaves, and this increase could play a crucial role in preserving the integrity of the plant cell membrane structure and enhancing the plant’s resistance against diseases. However, as the invasion prolonged, the plant’s autoimmune response became insufficient to counteract the pathogens, leading to a decline in its physiological indices and eventual death.

Pathogen-induced infection in plants could result in the occurrence of oxidative bursts, characterized by the rapid generation of reactive oxygen species (ROS) [38]. The excessive accumulation of ROS triggered an oxidative stress response in plants, leading to detrimental effects on nucleic acids, proteins, lipids, and other macromolecules and ultimately disrupting cellular function [39]. Plants developed various coping mechanisms to counteract the oxidative damage caused by the excessive ROS accumulation. An effective approach to detoxification involves enhancing the activity of antioxidant enzymes to safeguard plants [40]. Within plants, the key antioxidant enzymes include SOD, CAT, and POD, which served as a strategy to combat pathogen invasion [41]. SOD, functioning as the primary defense against oxidation, is a vital protective enzyme in plant tissues, primarily engaged in oxygen metabolism and contributing significantly to the investigation of plant disease resistance mechanisms [42]. Previous research demonstrated that disease-resistant varieties exhibited significantly higher SOD activity than susceptible varieties following infection by *F. trichothecioide* [43]. CAT, an essential protective enzyme in plant tissues, plays a crucial role in breaking down the accumulated H_2_O_2_ resulting from plant metabolism, thereby reducing oxidative damage to cells. Apart from its role as a prominent component of the antioxidant enzyme system, POD plays a crucial role in facilitating the synthesis of lignin, thereby promoting the process of lignification in affected tissues. It also acted as a catalyst for the production of phenols that possessed toxicity towards pathogens, effectively inhibiting their proliferation and expansion within the host organism. The antioxidant enzymes in plants consistently maintained a dynamic equilibrium with ROS, thereby regulating cellular levels of free radicals to a normal range. In this study, the activities of CAT and POD in the early stage of infection showed an upward trend, whereas the contents of ROS and free radicals did not change significantly, indicating that the outbreak of ROS activated the antioxidant system response in plants. With the prolonged duration of infection, the activities of CAT and POD exhibited a decline, whereas the concentration of H_2_O_2_ showed an accumulation, and the levels of MDA significantly increased. These observations suggested that during the middle and later stages of *F. solani* infection, the plants’ capacity to manage reactive oxygen diminished, leading to the accumulation of ROS within plants, which, in turn, could cause membrane lipid peroxidation and exacerbate cellular damage. This accumulation surpassed the regulatory limits of plant autoimmunity, leading to metabolic disturbances and disruption of normal physiological structure and function. Consequently, this process accelerates the senescence and demise of plant leaves.

During the course of pathogen infection, plants could augment the activity of defense enzymes to mitigate the accumulation of detrimental substances and bolster their own resistance [44]. Defense enzymes, such as PAL, PPO, and CHI [45], were notably instrumental in the synthesis of lignin and phenolic compounds, serving as barriers against pathogen invasion and mitigating toxic substances [46,47]. These enzymes could directly contribute to enhancing disease resistance by generating quinone substances that impede the growth and expansion of fungal hyphae. PAL is a crucial enzyme in the phenylpropane metabolic pathway in plants, and its activity serves as a dependable indicator for evaluating the disease resistance of host plants. PPO could reinforce plant disease resistance through its involvement in the synthesis of lignin precursors and the oxidation of phenolic substances in plants. CHI encodes a protein associated with disease progression, and it is capable of degrading the cell wall of pathogens and plays a significant role in plant defense mechanisms. During the initial phases of the disease, plants exhibited an augmented resistance mechanism through the reinforcement of enzymatic activity of PAL, PPO, and CHI to combat the pathogenic agents. However, as the disease progressed, the toxin accumulation in plants increased. Consequently, the innate resistance of the plants proved inadequate to counterbalance the hazardous substances imposed by the pathogen, ultimately resulting in withering and demise.

SA and JA are significant mediators of plant immunity against pathogens, with SA and JA signaling playing crucial roles in defending against biotrophic and necrotrophic pathogens [48]. Recent investigations shed light on the pivotal function of SA in impeding biotrophic infections and unraveled the biosynthesis and signaling pathways linked to the modification of pathogen susceptibility [49]. The results of the present study suggest that pathogen infection may trigger the activation of the SA defense pathway. However, additional research is needed to determine the extent of the involvement of the JA pathway in the defense response against *Fusarium* wilt in chrysanthemum, and related work is planned to be carried out in subsequent studies.

The soil environment is a multifaceted ecosystem, wherein the composition, diversity, and function of the soil microbial community are influenced by various factors such as climate, cultivation techniques, soil nutrient levels, introduction of invasive pathogens, and agricultural management practices. The soil microbial community plays a crucial role in suppressing soil-borne diseases through mechanisms such as promoting the synthesis of plant hormones, competing with soil-borne pathogens for essential nutrients, engaging in direct competition with plants, or activating immune responses regulated by microorganisms. The rhizosphere is widely recognized as the primary barrier against soil-borne pathogens. Plants have the ability to develop defense mechanisms against soil-borne pathogens through the targeted stimulation of antagonistic microorganisms, resulting in a significant increase in the abundance of these antagonistic microorganisms. In this study, the prolonged cultivation of chrysanthemum over numerous years led to the microbial community structure towards an unfavorable environment for plant growth. A significant decrease was observed in the proportion of beneficial bacteria, specifically *Nitrosospira* and *Pirellula*, and in the proportion of antagonistic microorganisms, such as Actinobacteria and *Aspergillus*. Conversely, a noteworthy increase was observed in the proportion of pathogen and saprophytic fungi, including species in *Myriococcum*, *Rhizophlyctis*, and *Ascobolus*. Moreover, a significant increase in the occurrence of *F. solani*, the causative agent of localized chrysanthemum wilt disease, was observed, surpassing a prevalence of 60%. Consequently, the soil microbial community underwent a gradual modification, adversely affecting plant growth and resulting in the emergence of soil-borne *Fusarium* wilt among the native chrysanthemum population. However, the specific molecular mechanisms governing the interaction between *F. solani* and chrysanthemum remain unexplored. Therefore, conducting further investigations on the interplay between pathogen and host is imperative because it could enhance the understanding of the fundamental molecular mechanisms implicated in pathogen infiltration. Choosing efficient soil-applied chemical fungicide, bio-fungicide, and bio-organic fertilizer, as well as their combined application, could be the best choice to improve rhizosphere microbial properties and effectively control the *Fusarium* wilt of chrysanthemums [50,51].

## Figures and Tables

**Figure 1 jof-10-00014-f001:**
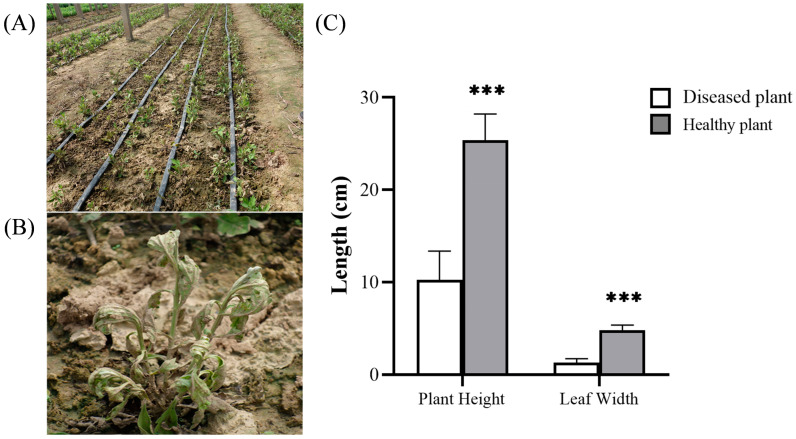
Chrysanthemum disease symptoms investigated in the field: (**A**) *Fusarium* wilt field. (**B**) Diseased plants. (**C**) Statistical analysis of plant height and leaf width of *Fusarium* wilt and healthy plants. ***: significant at 0.001 level.

**Figure 2 jof-10-00014-f002:**
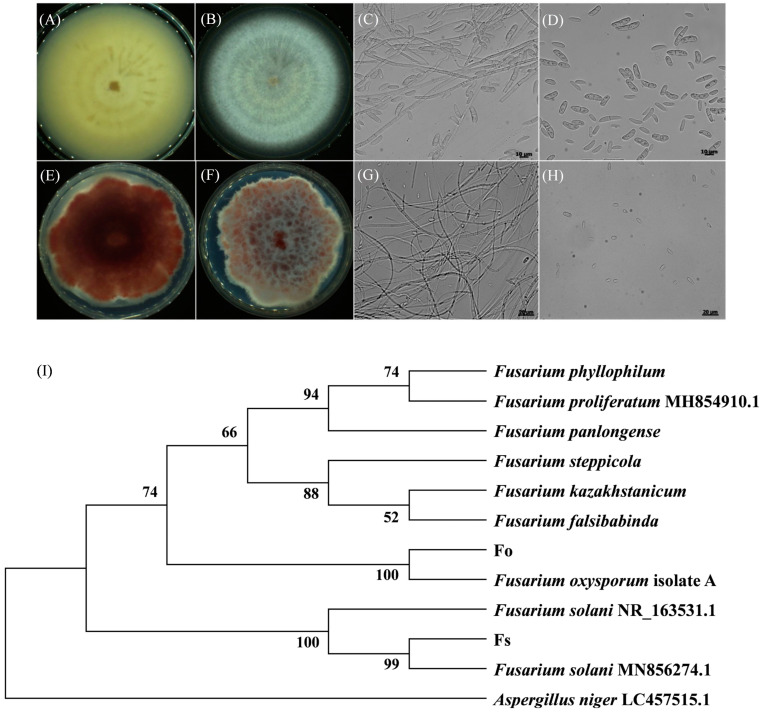
Morphological characteristics of *Fusarium* strains and identification: (**A**–**D**) Colony and light microscope images of type A strain at a magnification of 400×. (**E**–**H**) Colony and light microscope images of type B strain at a magnification of 400×. (**I**) Neighbor-joining tree constructed based on ITS sequences of strains *F. oxysporum* (Fo) and *F. solani* (Fs).

**Figure 3 jof-10-00014-f003:**
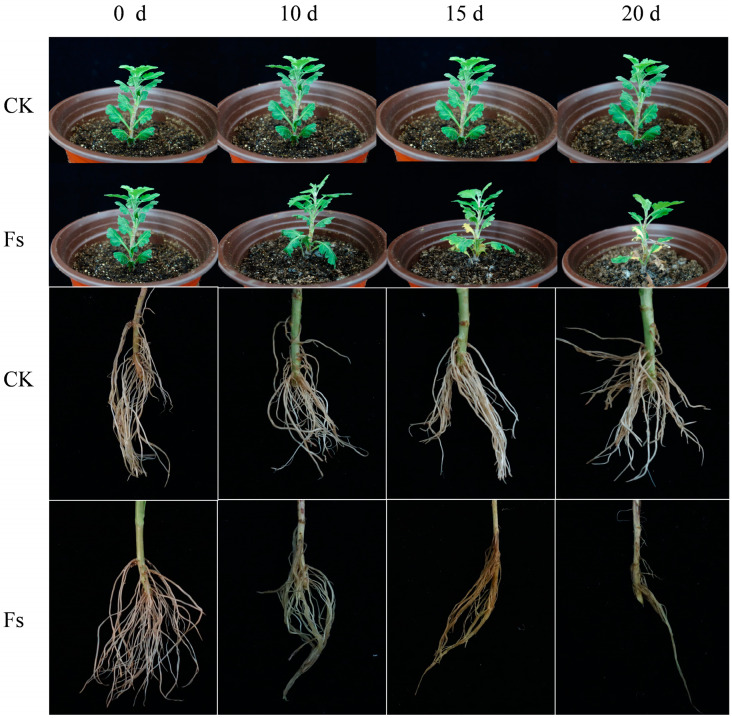
Pathogenic process of chrysanthemum following infection by *F. solani*: control group (CK); treatment group (Fs).

**Figure 4 jof-10-00014-f004:**
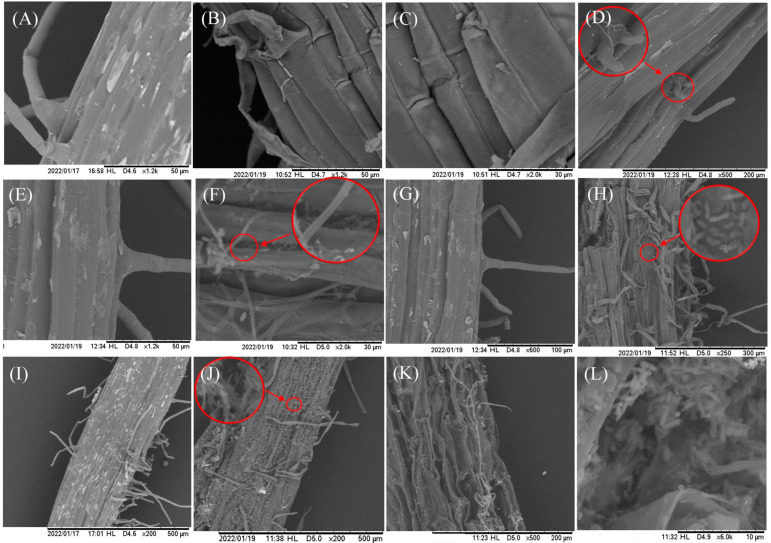
Invasion of chrysanthemum roots by *F. solani* under scanning electron microscopy (SEM) at magnifications ranging from ×200 to ×2000: (**A**,**C**,**E**,**G**,**I**) Roots of the control group observed at 12 h and 1, 3, 5, and 10 d. (**B**,**D**,**F**,**H**,**J**–**L**) Roots of the pathogen infection group observed at 12 h and 1, 3, 5, and 10 d.

**Figure 5 jof-10-00014-f005:**
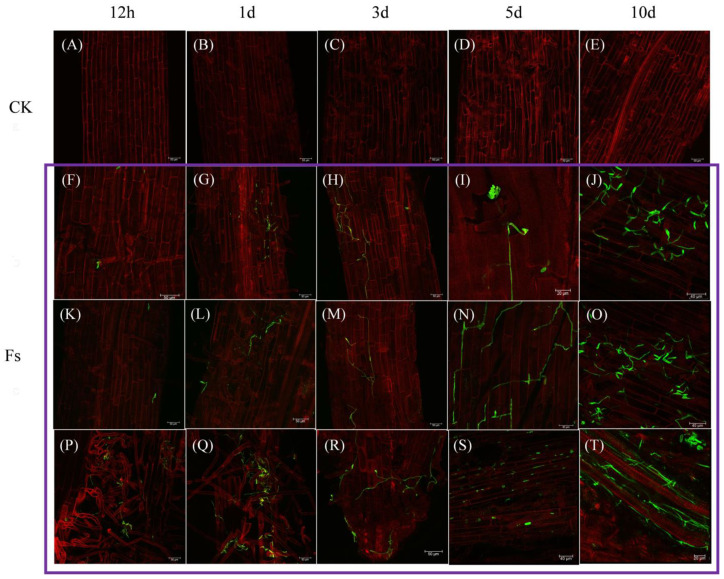
Process of *F. solani* infecting chrysanthemum roots under laser confocal scanning microscopy (LCSM, ×200–2000): (**A**–**E**) Roots of the control group at 12 h and 1, 3, 5, and 10 d. Purple square including (**F**,**K**,**P**), (**G**,**L**,**Q**), (**H**,**M**,**R**), (**I**,**N**,**S**), and (**J**,**O**,**T**) Roots of the pathogen infection group at 12 h and 1, 3, 5, and 10 d.

**Figure 6 jof-10-00014-f006:**
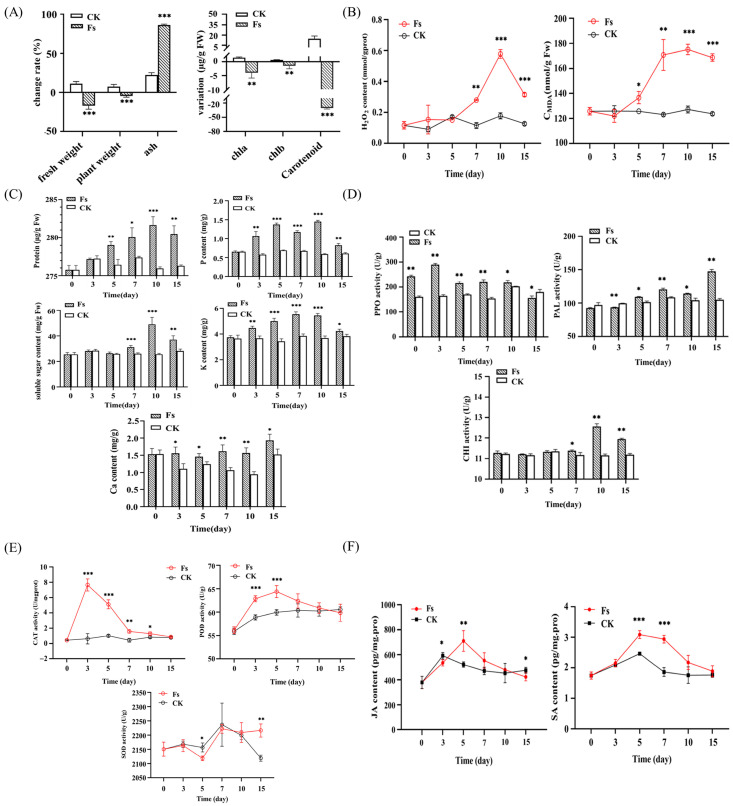
Physiological responses of chrysanthemum to *F. solani* inoculation: (**A**) Plant growth and photosynthetic pigment changes on 15 dpi of *F. solani*. (**B**) Changes in hydrogen peroxide and MDA content in plant leaves after *F. solani* inoculation. (**C**) Changes in nutrient contents in plants after inoculation by *F. solani*, including protein, soluble sugar, Ca, P, and K contents. (**D**) Changes in leaf defense enzyme activity after inoculation, including polyphenol oxidase, phenylalanine ammonialyase, and chitinase. (**E**) Changes in antioxidant enzyme activity in leaves after inoculation, including catalase, peroxidase, and superoxide dismutase. (**F**) Changes in JA and SA in leaves after *F. solani* inoculation. *: significant at 0.05 level, **: significant at 0.01 level, ***: significant at 0.001 level.

**Figure 7 jof-10-00014-f007:**
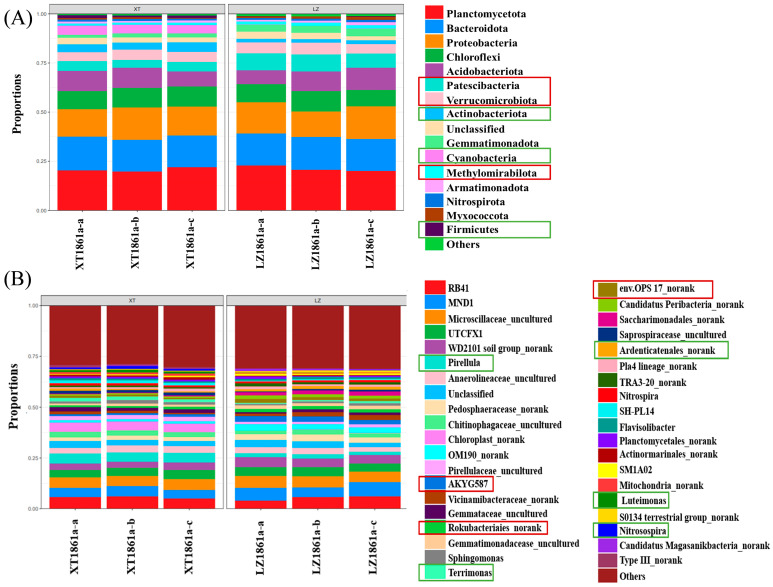
Analysis of soil bacterial microbial community structure: (**A**) Relative abundance of bacterial at the phylum level. (**B**) Relative abundance of bacteria at the genus level. Red square: the relative abundance of bacteria increased, Green square: the relative abundance of bacteria decreased (in continuous cropping soil compared with in healthy soil).

**Figure 8 jof-10-00014-f008:**
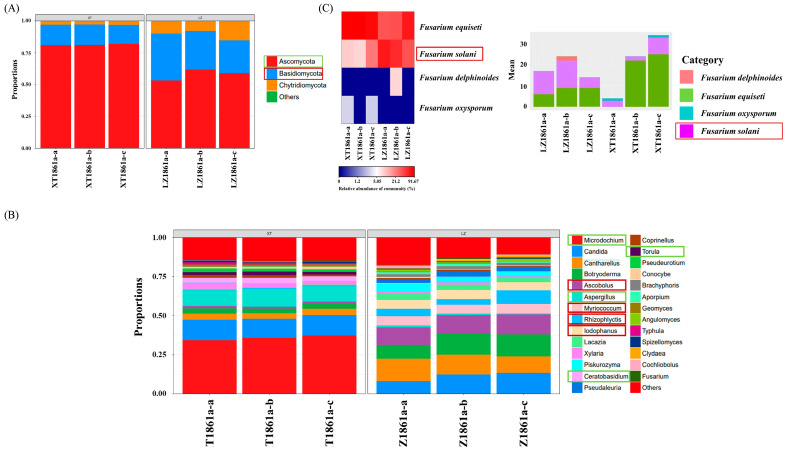
Analysis of soil fungal microbial community structure: (**A**) Relative abundance of fungi at the phylum level. (**B**) Relative abundance of fungi at the genus level. (**C**) Relative abundance of *Fusarium* at the species level. Red square: the relative abundance of fungi increased, Green square: the relative abundance of fungi decreased (in continuous cropping soil compared with in healthy soil).

## Data Availability

The datasets generated for this study can be found in the NCBI BioProject databases with access code PRJNA865142.

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
