# Peer review of "A Study of Soil-Borne Fusarium Wilt in Continuous Cropping Chrysanthemum Cultivar ‘Guangyu’ in Henan, China"

_jof, 2023, doi:10.3390/jof10010014_

Round 1

Reviewer 1 Report

Comments and Suggestions for Authors

I think that reference 3 probably deserves more discussion, since it deals with a very related topic. In addtion, two papers not cited in the manuscript would seem to be relevant: Chen, et al. 2018, Agriculture 8,184 and Zhao, et al. 2016, Molecules, 21,526. Results from these could be compared and contrasted to those presented in the manuscript.

Minor comments

Line 107: It would be best to indicate that the "healthy field" was a first-year cropping field, something like "...healthy field (first-year cropping)..."

Line 113: "multi-point sampling method" can be omitted, since it's made clear that's what was used in the rest of the sentence.

Line 207: instead of "rpm", use x g, or indicate the rotor used.

Fig. 3 legend: I think A, C, E is 0 days, B, D, F is 30 days: the legend needs to be corrected.

Line 334: "These enzymes played a crucial role in enabling F. solani to penetrate plant tissue." This is a supposition, not shown by the mere fact that the pathogen possesses these enzymes. In addition, have these enzyme activities been shown for Fusarium in other studies?

Line 368: meaning of "side space"?

Fig. 7 legend does not appear to match what's shown in the figure.

Fig. 8 and 9: can't read the text in the figures, need to be enlarged.

Line 551: "The findings indicated that the practice of continuous cropping resulted in a notable rise in the relative abundance of F. solani, aligning with previous research outcomes." A reference(s) to the "previous research outcomes" is needed.

Comments on the Quality of English Language

I have not made a list of changes or corrections: overall, the paper needs modification to correct some uses of tenses, spelling, etc.

Reviewer 2 Report

Comments and Suggestions for Authors

Dear colleagues!

The article “A study of soil-borne Fusarium wilt in continuous cropping Chrysanthemum cultivar 'Guangyu' in Henan, China” is aimed to study the mechanisms underlying wilt disease in the 'Guangyu' cultivar. A pathogenic organism (Fusarium solani) causing chrysanthemum disease has been detected, isolated and identified, and the activity of pathogen has been analyzed.

The list of comments:

1. The section “Introduction”. Line 34: “in continuous cropping areas”. The authors did not indicate what is happening in other cropping areas.

2. The section “Introduction”. Line 42-43: “second genome of plants”. Authors need to specify references.

3. The section “Introduction”. Lines 49-51: “However, in conducive soil environments, pathogens are able to rapidly invade and proliferate, consequently diminishing the presence of beneficial bacteria.”. Not necessarily that way. Incorrect phrase.

4. The section “Introduction”. Lines 51-54: it is unclear why it is Rehmannia glutinosa that is analyzed, and not plants of the Asteraceae family (genus Chrysanthemum).

5. The section “Introduction”. Lines 61-71: shorten the text.

6. Raw text in the section “Materials and methods”. The section “Materials and methods” and the section “Results” need to be significantly reduced. It is necessary to make the text of the sections more laconic and concise.

7. The section “Materials and methods”. Lines 106-108: “The continuous cropping field and healthy field of chrysanthemum were divided into five districts”. It is unclear the division of the two options into 5 districts. It is necessary to clarify.

8. The section “Materials and methods”. Lines 115-116: a reference or description of the rhizosphere soil sampling method are required.

9. The section “Materials and methods”. Lines 118-119: it is unclear whether samples were taken from the “healthy” field as a control.

10. The section “Materials and methods”. Line 135: “MEGA 4.0”. It is unclear why the old version of the program was used.

11. The section “Materials and methods”. Lines 137-147: shorten the text. Authors need to specify references.

12. The section “Materials and methods”. Lines 159-160: authors need to specify reference.

13. The section “Materials and methods”. Lines 158-191: shorten the text.

 14. The section “Materials and methods”. Lines 197-218: shorten the text.

15. The section “Materials and methods”. Lines 219-235: shorten the text, but need to detail “microbiome profiling”.

16. The section “Materials and methods”. Lines 249: it is unclear why there is no comparison with the NCBI database. Using the NCBI Genbank database is necessary.

17. The section “Materials and methods”. Line 204-214: there are no links to the sources of methods. Authors need to specify references.

18. The section “Materials and methods”. Line 215-218: there are no links to the sources of methods. Authors need to specify references.

19. The section “Results”. Figure 3 (A and E), Figure 7 (all texts in all figures), Figure 8 (B, C, D, E and F), Figure 9 (D), Figure 10 (A and E) are unreadable (low-contrast unfocused photos, unreadable text). The authors need to improve the quality of the figures. Move the figures 3, 4, 9A, 10A, table 1 to “Supplementary Materials” section or “Appendix”.

20. The section “Results”. Table 1 (Line 359): pathogenicity test of F. oxysporum showed in Figure 3, but in Table 1 there is no disease statistics for this treatment variant. Yes, F. oxysporum is a well-known object, but the authors obtained 37 strains of F. oxysporum and less, only 21 strains of F. solani. It is necessary to indicate good reasons for excluding F. oxysporum from further analysis.

21. The section “Results”. Lines 437-440: move to the “Discussion” section.

22. The section “Results”. Lines 534-535: it is unclear why only three divisions (“Ascomycota, Basidiomycota, and Chytridiomycota") are found. This is very little. In the “Discussion” section, authors need to specify similar cases, if there were any.

23. The section “Results”. Line 545: delete the phrase “, and other unnamed genera”. Unnamed genera cannot be discussed.

24. The “Discussion” section. Lines 567-574: this is not a discussion of the results, but a work plan. Remove.

25. The “Discussion” section. Lines 655-657: “However, additional research was needed to determine the extent of the involvement of the JA pathway in the defense response against Fusarium wilt in chrysanthemum.”. It is unclear whether this work was planned to be carried out, or whether the authors plan to carry out this research in the future.

26. The “Discussion” section. Lines 671-672: “Significant decrease was observed in the populations of beneficial bacteria”. Mistake. Only species richness with proportion reads/OTUs, and not quantity were evaluated.

27. Check the meaning of phrases and English translation: “delving into the underlying mechanisms” (Line 91); “raged” (Line 36); “root-dranching" (Line 149); “observed … to observe” (Lines 189-190); must be plural everywhere in “carotenoid and chlorophyll (chla, chlb), soluble sugar, and soluble protein” (Line 198-199); “200-2.0 k×. (A, C, E, G, I)” (may be “×200-2,000 (A, C, E, G, I).”; Line 378); “400×” (may be “×200-2,000”; Line 382); “Consequently” (may be “Thus”; Line 405); “pigment … carotenoid” (must be plural; Line 411); “findings” (Line 443); check the translation of the phrases “oxydative outbreak” (Lines 597-598), “rapid and instantaneous generation” (Line 598), “Rhizospheres … are” (may be a singular, Line 40); “relentless” (Line 646); “was needed”  (may be “is needed”; Line 656); “foreign” (may be “invasive”, Line 660).

Reviewer's conclusion: reconsider after major revision.

Comments on the Quality of English Language

Dear colleagues!

The article “A study of soil-borne Fusarium wilt in continuous cropping Chrysanthemum cultivar 'Guangyu' in Henan, China” is aimed to study the mechanisms underlying wilt disease in the 'Guangyu' cultivar. A pathogenic organism (Fusarium solani) causing chrysanthemum disease has been detected, isolated and identified, and the activity of pathogen has been analyzed.

The list of comments:

1. The section “Introduction”. Line 34: “in continuous cropping areas”. The authors did not indicate what is happening in other cropping areas.

2. The section “Introduction”. Line 42-43: “second genome of plants”. Authors need to specify references.

3. The section “Introduction”. Lines 49-51: “However, in conducive soil environments, pathogens are able to rapidly invade and proliferate, consequently diminishing the presence of beneficial bacteria.”. Not necessarily that way. Incorrect phrase.

4. The section “Introduction”. Lines 51-54: it is unclear why it is Rehmannia glutinosa that is analyzed, and not plants of the Asteraceae family (genus Chrysanthemum).

5. The section “Introduction”. Lines 61-71: shorten the text.

6. Raw text in the section “Materials and methods”. The section “Materials and methods” and the section “Results” need to be significantly reduced. It is necessary to make the text of the sections more laconic and concise.

7. The section “Materials and methods”. Lines 106-108: “The continuous cropping field and healthy field of chrysanthemum were divided into five districts”. It is unclear the division of the two options into 5 districts. It is necessary to clarify.

8. The section “Materials and methods”. Lines 115-116: a reference or description of the rhizosphere soil sampling method are required.

9. The section “Materials and methods”. Lines 118-119: it is unclear whether samples were taken from the “healthy” field as a control.

10. The section “Materials and methods”. Line 135: “MEGA 4.0”. It is unclear why the old version of the program was used.

11. The section “Materials and methods”. Lines 137-147: shorten the text. Authors need to specify references.

12. The section “Materials and methods”. Lines 159-160: authors need to specify reference.

13. The section “Materials and methods”. Lines 158-191: shorten the text.

 14. The section “Materials and methods”. Lines 197-218: shorten the text.

15. The section “Materials and methods”. Lines 219-235: shorten the text, but need to detail “microbiome profiling”.

16. The section “Materials and methods”. Lines 249: it is unclear why there is no comparison with the NCBI database. Using the NCBI Genbank database is necessary.

17. The section “Materials and methods”. Line 204-214: there are no links to the sources of methods. Authors need to specify references.

18. The section “Materials and methods”. Line 215-218: there are no links to the sources of methods. Authors need to specify references.

19. The section “Results”. Figure 3 (A and E), Figure 7 (all texts in all figures), Figure 8 (B, C, D, E and F), Figure 9 (D), Figure 10 (A and E) are unreadable (low-contrast unfocused photos, unreadable text). The authors need to improve the quality of the figures. Move the figures 3, 4, 9A, 10A, table 1 to “Supplementary Materials” section or “Appendix”.

20. The section “Results”. Table 1 (Line 359): pathogenicity test of F. oxysporum showed in Figure 3, but in Table 1 there is no disease statistics for this treatment variant. Yes, F. oxysporum is a well-known object, but the authors obtained 37 strains of F. oxysporum and less, only 21 strains of F. solani. It is necessary to indicate good reasons for excluding F. oxysporum from further analysis.

21. The section “Results”. Lines 437-440: move to the “Discussion” section.

22. The section “Results”. Lines 534-535: it is unclear why only three divisions (“Ascomycota, Basidiomycota, and Chytridiomycota") are found. This is very little. In the “Discussion” section, authors need to specify similar cases, if there were any.

23. The section “Results”. Line 545: delete the phrase “, and other unnamed genera”. Unnamed genera cannot be discussed.

24. The “Discussion” section. Lines 567-574: this is not a discussion of the results, but a work plan. Remove.

25. The “Discussion” section. Lines 655-657: “However, additional research was needed to determine the extent of the involvement of the JA pathway in the defense response against Fusarium wilt in chrysanthemum.”. It is unclear whether this work was planned to be carried out, or whether the authors plan to carry out this research in the future.

26. The “Discussion” section. Lines 671-672: “Significant decrease was observed in the populations of beneficial bacteria”. Mistake. Only species richness with proportion reads/OTUs, and not quantity were evaluated.

27. Check the meaning of phrases and English translation: “delving into the underlying mechanisms” (Line 91); “raged” (Line 36); “root-dranching" (Line 149); “observed … to observe” (Lines 189-190); must be plural everywhere in “carotenoid and chlorophyll (chla, chlb), soluble sugar, and soluble protein” (Line 198-199); “200-2.0 k×. (A, C, E, G, I)” (may be “×200-2,000 (A, C, E, G, I).”; Line 378); “400×” (may be “×200-2,000”; Line 382); “Consequently” (may be “Thus”; Line 405); “pigment … carotenoid” (must be plural; Line 411); “findings” (Line 443); check the translation of the phrases “oxydative outbreak” (Lines 597-598), “rapid and instantaneous generation” (Line 598), “Rhizospheres … are” (may be a singular, Line 40); “relentless” (Line 646); “was needed”  (may be “is needed”; Line 656); “foreign” (may be “invasive”, Line 660).

Reviewer's conclusion: reconsider after major revision.

Round 2

Reviewer 1 Report

Comments and Suggestions for Authors

The author's addressed all of the concerns and comments of the reviewers and the changes made are appropriate and sufficient. There's a misspelling of "fungus" in the title of Table S3.

Author Response

There's a misspelling of "fungus" in the title of Table S3.

Response:  Thank you very much for the error you mentioned, and the spelling of "fungus" has been corrected.

Reviewer 2 Report

Comments and Suggestions for Authors

Dear authors 

The manuscript were substantially revised. Comment: there is unreadable text in the Figures 7.A, S3.B, S4.B. The authors can improve the quality of the Figures. 

Expert opinion: accept after minor revision.

Comments on the Quality of English Language

The utilization of editing services has been employed to enhance the quality of the English language in accordance with specified criteria.

Author Response

There is unreadable text in the Figures 7.A, S3.B, S4.B. The authors can improve the quality of the Figures. 

Response: We sincerely appreciate your valuable advice. Text in the Figures 7.A, S3.B, S4.B has been improved, and supplementary attachments containing clearer images will be provided.